# Translating Genetic Discovery into a Mechanistic Understanding of Pediatric Movement Disorders: Lessons from Genetic Dystonias and Related Disorders

*Wei-Sheng Lin*

**The era of next-generation sequencing has increased the pace of gene discovery in the field of pediatric movement disorders. Following the identification of novel disease-causing genes, several studies have aimed to link the molecular and clinical aspects of these disorders. This perspective presents the developing stories of several childhood-onset movement disorders, including paroxysmal kinesigenic dyskinesia, myoclonus-dystonia syndrome, and other monogenic dystonias. These stories illustrate how gene discovery helps focus the research efforts of scientists trying to understand the mechanisms of disease. The genetic diagnosis of these clinical syndromes also helps clarify the associated phenotypic spectra and aids the search for additional disease-causing genes. Collectively, the findings of previous studies have led to increased recognition of the role of the cerebellum in the physiology and pathophysiology of motor control—a common theme in many pediatric movement disorders. To fully exploit the genetic information garnered in the clinical and research arenas, it is crucial that corresponding multi-omics analyses and functional studies also be performed at scale. Hopefully, these integrated efforts will provide us with a more comprehensive understanding of the genetic and neurobiological bases of movement disorders in childhood.**

## 1. Introduction

Movement disorders in childhood comprise a wide range of etiologically and clinically heterogeneous conditions that are often difficult to diagnose and treat. Investigations of pediatric movement disorders require a multidisciplinary approach that includes various modalities of neuroimaging, electrophysiology, and genetic and metabolic evaluations. With the advent of next-generation sequencing (NGS) technologies, the list of causative genes for pediatric movement disorders has expanded rapidly. As such, genetic testing has become increasingly important in the evaluation of children and adolescents with movement disorders.[1,2] Identifying the genetic etiology of pediatric movement disorders is not only clinically meaningful in therapeutic decision making and counseling,[3] but also refines our knowledge about these disorders, as exemplified by recent progress in several childhood-onset dystonia syndromes. Dystonia is defined as involuntary muscle contractions affecting one or more body parts, thereby causing twisting movements or abnormal postures. Several disorders featuring early-onset dystonia are known to be genetically determined. These dystonia syndromes can be classified based on their associated features; isolated dystonia refers to conditions in which dystonia is the sole movement disorder phenotype, whereas combined dystonia denotes conditions in which dystonia co-exists with other involuntary movements (such as myoclonus).[4] In addition, some dystonia syndromes are paroxysmal in nature; that is, they are characterized by episodic occurrences of involuntary movements.[5] This Perspective focuses on selected conditions representing paroxysmal and combined dystonias; namely, paroxysmal kinesigenic dyskinesia (PKD) and myoclonus-dystonia syndrome (MDS). In addition, isolated dystonias will also be discussed briefly. This review is not intended to be exhaustive; rather, it aims to showcase the winding road from gene discovery to mechanistic insights in this field. Taken together, these developing stories demonstrate how gene discovery inspires and anchors the efforts of basic and clinical investigations, which in turn help delineate the phenotypic spectra and pathophysiological mechanisms of pediatric movement disorders.

W.-S. Lin
Department of Pediatrics
Taipei Veterans General Hospital
Taipei 11217, Taiwan
E-mail: wslin7@vghtpe.gov.tw
W.-S. Lin
School of Medicine
National Yang Ming Chiao Tung University
Taipei 112304, Taiwan

iD The ORCID identification number(s) for the author(s) of this article can be found under https://doi.org/10.1002/ggn2.202200018

## 2. The Story of Paroxysmal Dyskinesia

### 2.1. The Discovery of *PRRT2*, the First PKD-Causing Gene

Paroxysmal dyskinesia is characterized by brief episodic attacks of dystonia and/or choreoathetotic movements. Both familial and sporadic cases have been identified, with most patients showing onset during childhood. Paroxysmal dyskinesia takes several forms that are named after their respective triggering factors and are assigned individual codes in the Online Mendelian Inheritance in Man (OMIM) database. These include PKD (OMIM #128200), paroxysmal non-kinesigenic dyskinesia (PNKD; OMIM #118800), and paroxysmal exercise-induced dyskinesia (also termed paroxysmal exertion-induced dyskinesia, PED; OMIM #612126).[6] PKD is the most common form of paroxysmal dyskinesia, and it is usually kinesigenic (that is, precipitated by sudden movements). This readily recognizable clinical picture of PKD, along with its exquisite responsiveness to low-dose sodium channel blockers, has been delineated several decades ago.[7–9] However, the electrophysiological underpinnings of PKD were a subject of debate.[10] PKD is sometimes associated with a family or personal history of self-limited infantile seizures, and this condition is known as PKD with infantile convulsions (PKD/IC, also termed infantile convulsions and paroxysmal choreoathetosis, ICCA; OMIM #602066).[11] At the turn of the century, linkage analysis was used to map the locus for these conditions to the pericentromeric region of chromosome 16; subsequently, similar results have been reported by multiple research groups.[11–16] Despite extensive attempts to identify the causative gene following the identification of the locus,[16] the gene responsible for PKD—the proline-rich transmembrane protein 2 gene (*PRRT2*)—was first identified in 2011 through exome sequencing.[17,18] *PRRT2* was subsequently confirmed as a major causative gene of PKD in multiple clinical cohorts of different racioethnic backgrounds, and it was also found to be associated with a variety of other neurological conditions, including epilepsy, migraine, and episodic ataxia.[19–21] Moreover, *PRRT2*-associated phenotypic heterogeneity was reported even within families.[22]

### 2.2. Recent Progress in the Mechanistic Understanding of *PRRT2*-related PKD

At the time when *PRRT2* was identified as the causative gene for PKD, very little was known about its functions at the cellular and neural circuit levels. In vitro studies have reported that disease-causing mutations result in decreased or absent protein expression. However, cells co-transfected with mutant and wild-type *PRRT2* did not show evidence of a dominant-negative effect, which suggested haploinsufficiency as the pathogenic mechanism.[23] This view was further supported by the identification of 16p11.2 microdeletions involving *PRRT2* in some patients.[24,25] However, not every mutant transcript is subject to nonsense-mediated decay,[26] and the PRRT2 protein is known to oligomerize,[27] suggesting that the dominant-negative effect may play a pathogenic role for some variants. Interestingly, the penetrance for paroxysmal movement disorder appears to be lower in cases with 16p11.2 deletions, which is a relatively common form of microdeletion syndrome associated with a vari-

ety of neurodevelopmental disorders.[28] It remains to be clarified whether this is due to incomplete characterization of the associated phenotypic features, or if the contiguous gene(s) has epistatic or other disease-modifying effects.

In line with findings from human patients, rodent models of PRRT2 deficiency also exhibit paroxysmal movement disorder phenotypes.[29,30] This makes them suitable models for investigating the mechanisms of these disorders. The various presynaptic functions of PRRT2 have been better understood in recent years. First, PRRT2 interacts with several presynaptic proteins involved in the synaptic vesicle cycle—notably SNAP-25 and vesicle-associated membrane protein 2 (VAMP2, also known as synaptobrevin-2)—to regulate the release of neurotransmitters.[29,31] Notably, mutations in *SNAP25* and *VAMP2* are associated with neurodevelopmental disorders with dystonia.[32] Second, PRRT2 interacts directly with P/Q-type calcium channels (encoded by *CACNA1A*), and both PRRT2 and CACNA1A are heavily expressed in the cerebellum.[33] PRRT2 deficiency leads to decreased membrane targeting of P/Q-type calcium channels, thus reducing the presynaptic calcium influx currents mediated by these channels.[33] Therefore, the significant overlap between *PRRT2*- and *CACNA1A*-related neurological phenotypes—including paroxysmal dyskinesia, episodic ataxia, hemiplegic migraine, and epilepsy—is likely non-fortuitous.[5,34] Third, proteomic studies showed that the $\alpha 3$ subunit of $Na^+/K^+$-ATPase (encoded by *ATP1A3*) is one of the major binding partners of PRRT2.[35] *ATP1A3* is associated with a wide spectrum of neurological phenotypes (which are sometimes complicated),[36] and both *PRRT2* and *ATP1A3* mutations are implicated in distinct forms of paroxysmal dyskinesia and epilepsy.[5] The phenotypic similarities suggest that the interference of their interactions may be related to the pathogenesis of these disorders.

PRRT2 has also been found to be expressed in lower amounts in the postsynaptic compartment.[29] Proteomic studies show that PRRT2 may act as an AMPA receptor accessory protein, suggesting that it is involved in postsynaptic function.[37] Further research is needed to elucidate the role of PRRT2 in this aspect. In addition, PRRT2 may also act at extra-synaptic sites by negatively modulating neuronal voltage-gated sodium channels, including $Na_v1.6$ (encoded by *SCN8A*).[38,39] In this regard, it is interesting to note that a specific form of *SCN8A* mutation (c.4447G>A), which presumably causes a slight gain of function, may also manifest as PKD/IC.[40] Both *PRRT2*- and *SCN8A*-related PKD are responsive to therapy with sodium channel blockers. However, the paroxysmal electroencephalographic changes recorded during dyskinetic attacks suggest that *SCN8A*-related PKD could be epileptic in nature (**Figure 1**).

Taken together, the diverse subcellular localizations (presynaptic, postsynaptic, and extra-synaptic) and multitude of interacting partners of PRRT2 indicate that it is a multifunctional or multitasking molecule. Therefore, *PRRT2* variants may have complicated and likely variant-specific impacts at the level of the neural circuit. To further dissect the mechanistic basis of *PRRT2*-related PKD, it is important to clarify the principal neural substrates that mediate this phenomenology. PRRT2 is highly expressed in the cerebellar cortex—specifically in granule cells[41]—and knockout *Prrt2* in cerebellar granule cells is sufficient to induce paroxysmal dyskinetic attacks in mice.[31,38] Furthermore, aberrant cerebellar outputs have been causally

**Table 1.** A comparison of *PRRT2*- and *TMEM151A*-related paroxysmal kinesigenic dyskinesia.

| | PRRT2 | TMEM151A |
|---|---|---|
| Mode of inheritance | Autosomal dominant | Autosomal dominant |
| Familial/sporadic cases | +/+ | +/+ |
| Penetrance[a] | ≈61%–89%[57,58] | 7/13[46] |
| Rate of de novo mutation | ≈5.5% | To be determined (a few cases were reported)[46,59] |
| Movement disorder features | | |
| ○ Attack duration | Longer | Shorter |
| ○ Phenomenology | Choreoathetosis predominant | Dystonia predominant |
| ○ Clinical response to sodium channel blockers | Yes, often complete | Yes, often incomplete |
| Association with benign infantile epilepsy | + | - or rare[60] |
| Mutation effect[b] | LOF, haploinsufficiency | LOF, haploinsufficiency (limited data)[44] |
| Animal studies | Mouse model,[30,31] | Mouse model[44] |
| | Rat model[29] | |

[a] Penetrance with regard to PKD phenotype; [b] Abbreviation: +, presence; -, absence; LOF, loss of function.

linked to paroxysmal dyskinetic attacks in these animals.[38] In line with the findings from animal models, a recent study found evidence for structural alterations in both cerebellar gray and white matter. In addition, functional magnetic resonance imaging studies have reported abnormal effective connectivity between the cerebellum and other nodes of the motor control network in patients with *PRRT2*-related PKD.[42]

## 2.3. Genetic Heterogeneity and Gene–Phenotype Correlations in Paroxysmal Dyskinesia

Over the past decade, *PRRT2* is the only gene that has consistently been found to be related to PKD. Variants of several other genes (such as *SCN8A*, as discussed above) have been suggested to be responsible for a few cases of PKD.[5,43] However, the genetic etiology of a significant fraction of patients with *PRRT2*-negative familial or sporadic PKD remains to be identified, suggesting the presence of other causative gene(s). Whole exome sequencing studies have recently revealed that *TMEM151A* variants are responsible for PKD in three dominantly-inherited families and in several sporadic cases.[44] *TMEM151A*-related PKD resembles the phenotype associated with *PRRT2*,[45] although subtle differences between them have been noted (**Table 1**).[46] Despite being highly conserved across species, the cellular function of *TMEM151A* has been rarely investigated and remains virtually unknown at present. Similar to *Prrt2*, *Tmem151a* is highly expressed in the brain and spinal cord,[44] and *Tmem151a* knockout mice recapitulate the movement disorder phenotype observed in human patients.[44] With these recent advances, it is expected that the role of *TMEM151A* in neural circuit operations and motor control will be gradually elucidated in coming years.

Childhood-onset paroxysmal dyskinesia often exhibits age-dependent evolution of phenotypic features. For example, PKD usually worsens during adolescence and young-adult life and then gradually improves—or even resolves—at a later age.[47] Multiple mechanisms may underlie the course of the disorder, including physiological brain maturation, adaptive and aberrant plasticity, and age-dependent gene expression.[48,49] Investigations into the developmental stage-dependent expression patterns of *Prrt2* and *Tmem151a* in mice have offered insights into the age-dependent expression of these genes. In particular, *Prrt2* knockout mice have been shown to recapitulate the age dependence of dyskinesia in human patients.[44,50]

The search for genes associated with paroxysmal dyskinesia has provided insights into the *TBC1D24* gene. At the end of the last century, Guerrini et al. performed linkage analysis on a pedigree with three members having self-limited focal epilepsy, PED, and writer's cramp (OMIM #608105).[51] A common homozygous haplotype mapped to the chromosomal region 16p12-11.2 was found to co-segregate with the disease status. This locus overlapped with the previously identified PKD/IC locus, leading to the speculation that a common causative gene underlies these seemingly similar neurological phenotypes.[51] Due to the presence of multiple loops of consanguinity in the pedigree, the critical region was initially analyzed using a recessive consanguineous model; however, the results did not yield any potentially disease-causing homozygous variants in a consistent manner.[52] In a study that assumed a recessive model that included compound heterozygous mutations, the researchers identified *TBC1D24*—rather than *PRRT2*—as the causative gene for the disorder.[52] In retrospect, the constellation of neurological phenotypes described by Guerrini et al. is distinct from that of *PRRT2*-related disorders, and their inheritance pattern is also different. However, given that the genes implicated in movement disorders often have incomplete penetrance and variable expressivity, the distinction between different genetically-defined clinical entities could only be appreciated in hindsight.[53]

Gene discovery in the context of paroxysmal dyskinesia has not only prompted investigations into the neurobiological basis of paroxysmal movement disorders, but has also provided tantalizing hints with regard to the precipitating factors that often trigger or aggravate them. For instance, PNKD is often precipitated by exposure to alcohol, caffeine, stress, and fatigue. The main causative gene, *PNKD* (formerly known as *MR-1*), is homologous to the hydroxyacylglutathione hydrolase (*HAGH*) gene, with their protein products sharing b-lactamase domains.[54] HAGH detoxifies methylglyoxal, which is a compound present in coffee and alcohol and produced as a by-product of oxidative stress. Therefore, it was once speculated that PNKD and HAGH have similar

**Table 2.** A comparison between *SGCE*-, *KCTD17*-, and *KCNN2*-related MDS.

| | SGCE | KCTD17 | KCNN2 |
|---|---|---|---|
| Inherited/de novo | +/+ | +/+ | +/+ |
| Mode of inheritance | Autosomal dominant | Autosomal dominant | Autosomal dominant |
| Mutation effect | LOF | LOF | LOF (limited data)[89] |
| Phenotypic features | | | |
| ○ myoclonus | + | +, milder | +, more distal[86] |
| ○ dystonia | +, self-limited | +, progressive | + |
| ○ psychiatric disturbances | + | +/- (less as compared to *SGCE*-related MDS)[79] | less? (limited data; anxiety reported in a case)[86] |
| ○ alcohol responsiveness | + | - | -a) |
| Developmental delay | - | +(some, associated with splice-site mutations?)[81,82] | +(some; limited data)[87] |
| Responsiveness to pallidal DBS | + | + | Not reported |

a) Early alcohol responsiveness was noted in a case, which was lost over time;[86]  b) Abbreviations: +, presence; -, absence; MDS, myoclonus-dystonia syndrome; DBS, deep brain stimulation; LOF, loss of function.

enzymatic activities. Although this possibility was refuted in later experiments, PNKD may still have alternative catalytic functions related to potential triggering factors.[55] Interestingly, *tottering* mice—generated through a loss-of-function mutation of *Cacna1a* (an interacting partner of prrt2, see Section 2.2.)—also manifest paroxysmal dystonia triggered by stress, ethanol, and caffeine.[56] The shared precipitating factors between genetic forms of paroxysmal dyskinesia may have mechanistic implications worth further elucidation.

## 3. The Story of Myoclonus-Dystonia Syndrome

### 3.1. Gene Discovery and the Development of a Mechanistic Understanding of MDS

Myoclonus-dystonia syndrome (MDS; OMIM #159900) is characterized by dystonia and myoclonic jerks mainly involving the neck, trunk, and upper limbs. The myoclonus is generally more prominent and predominant over the dystonia. Other characteristic—but not universal—features of MDS include psychiatric disturbance and alcohol responsiveness. This constellation of clinical features has been noted for several decades.[61,62] The first locus for MDS was mapped to the chromosomal region 7q21-q31 more than 20 years ago.[63–66] Sequencing the genes within this locus yielded the first causative gene, epsilon-sarcoglycan (*SGCE*), whose heterozygous loss-of-function mutations segregate with MDS.[67] *SGCE*-related MDS is typically inherited from the father, and there is a marked difference in the penetrance depending on the parental origin of the disease allele. This finding led to the speculation of a maternal imprinting mechanism.[67] It was subsequently confirmed that the maternally-derived allele of *SGCE* is silenced by promoter methylation[68,69] and that the imprinting pattern is maintained in the human brain.[70]

Multiple isoforms of SGCE, including a brain-specific one (exon 11b), are expressed in humans.[70] Different isoforms have also been identified in pre- and postsynaptic compartments.[71] This multitude of isoforms suggests the complex roles of SGCE in neuronal biology, and a variety of cellular processes—such as calcium homeostasis, membrane stabilization, and cell cy-

cle control—have been implicated in its functioning.[72,73] Despite these findings, the relevance of these isoforms to the pathogenesis of MDS remains obscure. *SGCE* is highly expressed in the cerebellum in mice and humans.[70,71] Neuroimaging studies have also pointed to a phenotype-related metabolic change in the parasagittal cerebellum in individuals with *SGCE* mutations.[74] Eyeblink conditioning (a paradigm of cerebellar-dependent associative learning) is impaired in patients with *SGCE*-related MDS, but not in those with isolated dystonia.[75,76] In this paradigm, alcohol negatively affects performance in healthy individuals and improves it in patients with *SGCE*-related MDS. Taken together, the molecular, anatomic, and functional data converge on the pivotal role of the cerebellum in the pathogenesis of MDS. However, a study reported that cerebellar Purkinje cell-specific *Sgce*-knockout mice did not exhibit the cardinal features of MDS, suggesting the involvement of other brain regions.[77] In contrast, adult mice with acute cerebellar *Sgce* knockdown showed the salient features of MDS, including myoclonus, dystonia, and alcohol responsiveness.[78] This discrepancy suggests the existence of some compensatory mechanisms in the previous model.[73]

### 3.2. Genetic Heterogeneity in MDS: The Story Continues

Overall, *SGCE* mutations have been found in only roughly one-third to half of all patients with MDS, suggesting the presence of additional causative genes for this condition. An investigation of a dominantly-inherited British pedigree with eight affected subjects yielded a novel causative gene, potassium channel tetramerization domain-containing 17 (*KCTD17*).[79] This was achieved through linkage analysis coupled with exome sequencing (a common strategy used in the NGS era) to identify novel dystonia-causing genes.[80] The role of *KCTD17* in MDS was further corroborated by two independent reports of cases with de novo splice-site mutations in this gene.[81,82] However, *KCTD17*-related MDS differs from that caused by *SGCE* mutations. Specifically, the dystonia is predominant and progressive in *KCTD17*-related MDS, whereas it is usually non-progressive in *SGCE*-related MDS (**Table 2**).

In the past decade, several other genes have also been reported to be causative of MDS, although they were either challenged by later studies (e.g., *CACNA1B*) or are pending replication and validation (e.g., *RELN*).[80,83,84] Other genes, such as *SCN8A*, are associated with myoclonus and dystonia in some cases;[85] however, their overall presentations do not fit the typical clinical picture of MDS. Although some researchers have considered *SCN8A* to be a candidate gene for MDS, deleterious variants of this gene have not been found to date.[86]

A recent study analyzed a three-generation pedigree with dominantly inherited MDS and identified *KCNN2* as another causative gene.[86] After excluding the established genetic causes of MDS (*SGCE* and *KCTD17*) and several other potentially related genes (including *SCN8A* and *RELN*), a heterozygous missense variant of *KCNN2* was found to segregate with disease status. This variant results in the substitution of an amino acid residue that is evolutionarily conserved. A sporadic case of MDS with de novo mutation of *KCNN2* has also been reported.[87] *KCNN2* encodes the SK2 subunit of the small-conductance calcium-activated potassium (SK) channel, which mediates repolarizing/hyperpolarizing currents in some neurons in the central nervous system.[88] *KCNN2* is highly expressed in the cerebellum, and some patients harboring *KCNN2* variants exhibit cerebellar eye signs[86] or cerebellar ataxia.[89] Collectively, the discovery of *KCNN2*-related MDS reinforces the view that the cerebellum is involved in the pathogenesis of MDS. Various *Kcnn2* rodent models have been in use since before human mutations were identified. The Trdk rat is dominantly transmitted and mimics essential tremor in humans;[90] the *frissonnant* mouse is recessively inherited, and this was previously characterized as a model of Parkinsonism.[91,92] Re-examining these mutant animals may help explain the phenotypes of the disorder and their neurobiological bases in light of the findings from human patients.

## 4. Common Themes and Prospects

### 4.1. "Localization" in Genetic Movement Disorders in Childhood

The localization of pertinent neural substrates and circuits is a classical subject of interest in clinical neurology. Pediatric movement disorders, and especially hyperkinetic involuntary movements, were traditionally viewed as disorders involving mainly the basal ganglia circuitry.[93] Nevertheless, one of the converging themes in PKD and MDS (as reviewed in Sections 2.2 and 3.1) is the fundamental role of cerebellar circuits in the genesis of abnormal motor patterns. Indeed, both the cerebellum and the basal ganglia are phylogenetically old brain regions, and their reciprocal connections have been increasingly recognized to play important roles in motor control.[48,94] Reduced microstructural integrity has been observed in the cerebellar outflow tracts of patients with *TOR1A*-related dystonia (also known as DYT1 dystonia),[95] which is the prototypical and most common form of early-onset isolated dystonia.[96] Altered cerebellar synaptogenesis and cerebellar microstructural defects (particularly those involving Purkinje cells) have also been observed in mouse models of DYT1 dystonia.[97,98] Interestingly, torsinA—the protein product of *TOR1A*—modulates the degradation of misfolded epsilon-sarcoglycan,[99] suggesting a potential pathophysiological connection between DYT1 dystonia and *SGCE*-related MDS. The

*tottering* mice (harboring a *Cacna1a* mutation) exhibit paroxysmal dystonia that is associated with electrophysiological activities presumably generated within the cerebellar cortex,[100] and the ablation of cerebellar Purkinje cells in these mice results in the disappearance of dystonia.[101] Taken together, these findings lend further credence to the role of the cerebellum in genetic movement disorders. In contrast, a recent study investigated the expression and co-expression of dystonia-associated genes and found that they were enriched in the basal ganglia, and not in the cerebellum.[102] However, it should be noted that the study used primarily adult-derived data for analysis. As such, the discrepancy seems to suggest that in some genetic movement disorders, the cerebellum may play a more pivotal role in children than in adults.

Identifying the neural circuits or substrates involved in a particular movement disorder not only improves our mechanistic understanding of these disorders, but may also inform targeted interrogation and the rational design of therapeutic trials with deep brain stimulation (DBS) or transcranial magnetic stimulation (TMS).[103] For instance, a proof-of-concept trial (ClinicalTrials.gov identifier: NCT03481491) of cerebellar TMS showed promising results in patients with *PRRT2*-related PKD, corroborating the pivotal role of the cerebellum in this condition.[42] Collectively, the mechanistic insights inspired by genetic discoveries in pediatric movement disorders reinforce the notion that questions about motor control should be reframed to consider the basal ganglia–cerebellar–cerebral cortical network as a whole.[48,94]

### 4.2. Electrophysiological Aspects of Pediatric Movement Disorders

Information processing in the nervous system mainly relies on electrical signaling. Therefore, it is not unexpected that monogenic childhood movement disorders, as exemplified by *PRRT2*-related PKD and *SGCE*-related MDS, are associated with electrophysiological abnormalities.[76,104] A recent study demonstrated that mice with *PRRT2* deletion in cerebellar granule cells are susceptible to the occurrence of a slowly propagating wave of depolarization in the cerebellar cortex (including granule cells and Purkinje cells). This wave precedes the aberrant outputs of deep cerebellar nuclei that are time-locked to paroxysmal dyskinetic attacks.[38] The spreading depolarization in the cerebellar cortex is reminiscent of an electrophysiologically similar phenomenon in the cerebral cortex known as cortical spreading depression (CSD), which is a slowly propagating wave of neuronal and glial depolarization followed by sustained underactivity. Although CSD is the putative electrophysiological process responsible for the generation of migraine aura, patients with *PRRT2*-related PKD also often report auras (or a sense of impending attacks) prior to dyskinetic events. Moreover, migraine is one of the *PRRT2*-related phenotypes,[26] even though *PRRT2* variants are considered to be risk-modifying rather than disease-causing for migraine.[105] Altered functions of several interacting partners of PRRT2—such as the $\alpha 3$ isoform of Na$^+$/K$^+$-ATPase (encoded by *Atp1a3*) and the Ca$_V$2.1 calcium channel (encoded by *Cacna1a*)—have been associated with changes in the susceptibility to spreading depolarization in various brain regions.[106,107] As

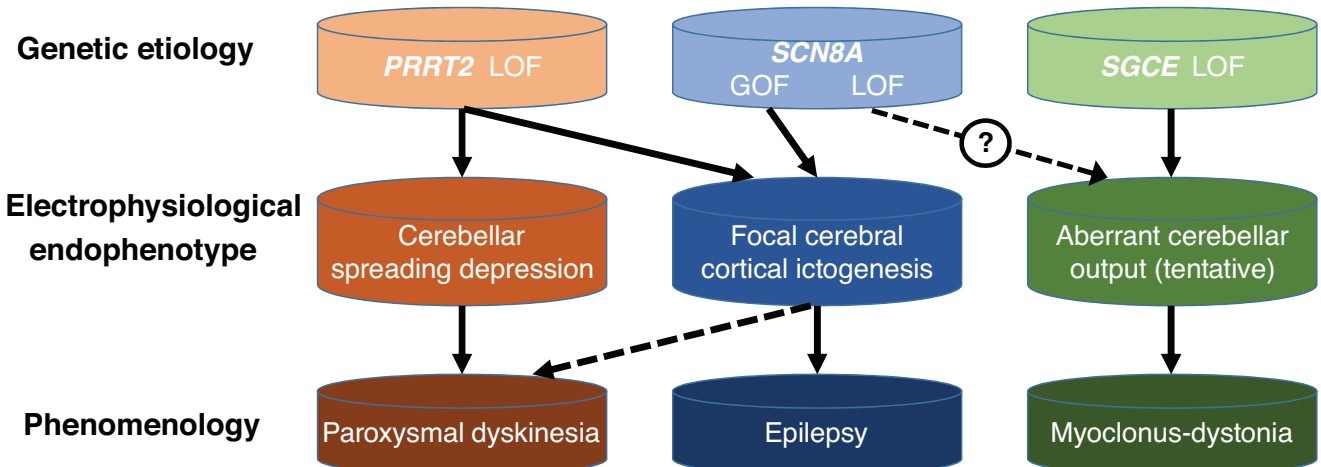

**Figure 1.** A three-pronged approach to the classification and characterization of pediatric movement disorders. The interposition of (tentative) endophenotypes between genetic etiology and phenomenology helps elucidate the complicated and interwoven mechanistic relationships—whether diverging or converging—across these three levels. Details are discussed in the relevant parts of the text. The dashed lines indicate associations with comparatively limited evidence available at the time of publication. GOF, gain of function; LOF, loss of function.

reviewed earlier (Section 2.2), these proteins are also associated with a variety of paroxysmal movement disorders. Taken together, the aforementioned findings raise the possibility that different neurological phenotypes may share similar electrophysiological underpinnings, leading to various manifestations depending on the specific neural substrates or circuits involved. It remains to be investigated whether other genes implicated in paroxysmal dyskinesia—such as *TMEM151A, SCN8A,* and *PNKD* (see Sections 2.2 and 2.3)—also predispose individuals to the spreading depolarization in the cerebellum.

Together with the circuit mapping discussed in Section 4.1, electrophysiological characterization of these disorders would contribute to the precise parameterization and effective delivery of treatment modalities such as DBS or TMS. It is hoped that developments along this direction would enable precise and personalized neuromodulation for selected pediatric movement disorders in the near future.

### 4.3. Genotype–Phenotype Correlations and the Transdiagnostic Approach

Genotype–phenotype relationships are often complex and incompletely understood in genetic movement disorders such as *PRRT2*-related disorders[57,108] and *SGCE*-related MDS.[109] A common bottleneck is the limited number of cases available for the study of each genetically-defined condition. There may also be a publication bias toward unusual manifestations of these disorders, resulting in the overrepresentation of atypical cases in the literature.[110,111] Nevertheless, the advent of the NGS era can help improve the rapid identification of patients with genetically-confirmed diagnoses. Three-digit case series are likely to be achieved for several forms of monogenic movement disorders, as has already occurred in the field of epilepsy genetics.[112] These are preferably achieved through prospective patient registries, which facilitate comprehensive phenotyping, standardized data collection, and longitudinal follow-up. In the long run, analyses

of large case series will provide a more balanced picture of phenotypic spectra and will yield more reliable genotype–phenotype correlations.[96]

Phenotypic variability is common in monogenic movement disorders in childhood. Moreover, various forms of involuntary movements often occur in combination and/or are embedded in more complex neurological scenarios, which may encompass epilepsy, neurodevelopmental disorders, and psychiatric disturbances.[3,89,112] It is widely assumed that variant-specific effects are one of the sources of phenotypic heterogeneity for a given monogenic movement disorder. Therefore, functional characterization of the neurobiological effects of each individual variant is a prerequisite for the development of precision medicine. However, this task is usually time-consuming and resource-intensive. Fortunately, with the development and systematic use of multi-omics and high-throughput laboratory assays (such as multiplexed assays of variant effect[113] and high-throughput electrophysiology[114]), variant-level functional assessments have now become more tractable.

Even if variant-level functional characterization is achieved, patients with the same genetic variants could still have disparate manifestations. For example, marked intrafamilial phenotypic variability has been reported in *PRRT2*- and *SGCE*-related disorders.[22,115] It is possible that other genetic factors also modify the expression of phenotypic features. The roles of nearby genes are particularly intriguing, as movement disorders seem to be less reported in conditions involving both causative and contiguous genes (e.g., PKD in the context of 16p11.2 microdeletion[25,116] and MDS in the context of Russell–Silver syndrome due to maternal uniparental disomy 7). More research is needed to clarify whether modifier genes really exist, or whether involuntary movements were just overlooked because of more complicated clinical pictures. An aspirational goal envisaged for human genomics states that by 2030, 'the clinical relevance of all encountered genomic variants will be readily predictable.'[117] However, the incomplete penetrance and variable expressivity of many dystonia-related genes

**Table 3.** A comparison of major advances in our genetic and mechanistic understanding of PKD, MDS, and pediatric-onset isolated dystonias.

|  | PKD | MDS | Isolated dystonias |
|---|---|---|---|
| Characterization of clinical features | Many decades ago[8,9] | Decades ago[61] | A century ago |
| Mode of inheritance | Autosomal dominant | Autosomal dominant | Dominant or recessive |
| First locus identified | 16p-q (1997)[14] | 7q (1999)[66] | 9q (1990)[119] |
| First causative gene identified | *PRRT2* (2011) | *SGCE* (2001) | *TOR1A* (1997) |
| Genotype-phenotype correlation studies | *PRRT2*[57] | *SGCE*[120] (no clear correlation) | *TOR1A*[96] |
| Complex genetic syndromes involving the disease-causing genes | 16p11.2 microdeletion[25,116] | Russell-Silver syndrome due to maternal uniparental disomy 7[121] | 9q34 deletion[122] |
| Animal models | + (mouse, rat) | + (mouse) | + (mouse, fruit fly) |
| Tissue distribution | Brain and spinal cord | Ubiquitous; with brain-specific isoform (exon 11b)[70] | Ubiquitous |
| The cerebellar cell type in which the gene is mainly expressed | Granule cells[41] | Purkinje cells and neurons of the dentate nucleus[70] | Granule cells, Purkinje cells, and the dentate nucleus[123,124] |
| Other possible causative genes discovered[a] | *TMEM151A* (2021) | *KCTD17* (2015), *CACNA1B* (2015), *RELN* (2015), *KCNN2* (2020) | *PRKRA* (2008), *THAP1* (2009), *HPCA* (2015), *KMT2B* (2016) |

[a] Some may need replication and/or validation. [b] The references cited in this table are meant to serve as examples and are not exhaustive. [c] Abbreviations: PKD, paroxysmal kinesigenic dyskinesia; MDS, myoclonus-dystonia syndrome.

make this a formidable goal in the field of pediatric movement disorders.[53]

Despite the complexity of deciphering genomic information, mutations in different genes often lead to a limited repertoire of clinical syndromes (such as PKD and MDS) in this field, suggesting the convergence of several pathogenetic pathways. Therefore, the transdiagnostic approach can provide an inroad into this body of knowledge. This can be achieved by using the electrophysiological characteristics of these disorders (as discussed in Section 4.2) as the endophenotypes (Figure 1). This approach expands on the current two-pronged naming system (i.e., phenotype prefix and gene name suffix, such as DYT-SGCE)[118] by inserting the endophenotype in between the genetic etiology and phenomenological classification, so that the diverse relationships between genes and phenotypes can be better appreciated. Moreover, the incorporation of endophenotypes—be it electroencephalographic signatures or other characteristics—highlights the complex pathophysiological processes linking genotypes and phenotypes. For example, both *PRRT2* and *SCN8A* mutations may present with paroxysmal dyskinesia, albeit through different mechanisms (see Section 2.2). Conversely, different genetic variants may lead to shared neuronal or circuit mechanisms, thereby culminating in similar phenomenology. For the sake of clarity, Figure 1 is somewhat oversimplified in some aspects of functional and phenotypic pleiotropy. Because different variants of a single gene can exert differential effects on multiple neural circuits serving diverse functions, these may as well be assimilated into this conceptual framework.

## 5. Conclusions

Thanks to advances in NGS technologies, the past decade has witnessed an acceleration in the pace of discovery of novel genes pertinent to pediatric movement disorders (**Table 3**). Indeed, several

disease-causing genes discussed in this article have not yet been cataloged in OMIM database and the Movement Disorder Society Genetic mutation database (MDSGene, https://www.mdsgene.org/; last accessed March 31, 2022). The discovery of these novel genes would only mark the end of the initial stages of our collective efforts in understanding the genetic bases of these movement disorders. As exemplified by the developing stories of PKD and MDS, gene discovery often helps anchor subsequent basic and clinical studies, which in turn elucidate the pathophysiology of respective clinical phenomena in more detail (from molecular mechanisms to the level of neural circuits). The iterative use of reverse genetics and forward genetics in human and animal models not only yields insights into disease mechanisms, but also facilitates the discovery of novel disease-causing or disease-predisposing genes. Ultimately, this strategy can help refine our understanding of the genetic landscape and pathophysiology of pediatric movement disorders.

## Supporting Information

Supporting Information is available from the Wiley Online Library or from the author.

## Conflict of Interest

The author declares no conflict of interest.

## Peer Review

The peer review history for this article is available in the Supporting Information for this article.

## Keywords

cerebellum, dystonia, myoclonus-dystonia syndrome, paroxysmal dyskinesia, pediatric movement disorders

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
