## [**Supplementary Information**: Record of Transparent Peer Review · Advanced Genetics]

**Translating genetic discovery into a mechanistic understanding of pediatric movement disorders:
Lessons from genetic dystonias and related disorders**

Wei-Sheng Lin*

* Corresponding author

Date submitted: 30 April 2022

Editor: Kerstin Brachhold, Andrew L. Hufton

1 st Peer Review Decision	02 June 2022
--------------

Dear Professor Lin,

Thank you for submitting your manuscript entitled "Translating genetic discovery into mechanistic understanding of pediatric movement disorders" (Perspective, No. ggn2.202200018) to Advanced Genetics. The reviewer report and comments are included at the end of this e-mail.

On the basis of these reviewer comments, we are not able at this stage to accept your manuscript for publication. I invite you to address the reviewer comments and make the necessary changes and improvements in a major revision of your manuscript.

Your manuscript also requires revision with respect to the language used. We therefore suggest that you ask a native English speaker or equivalent to assist you with correcting the spelling, grammar, word use, and punctuation throughout your manuscript. To help with this process you may wish to contact a professional language editing service, such as Wiley Editing Services (https://wileyeditingservices.com/en/article-preparation/english-language-editing?utm_source=email&utm_medium=referral&utm_term=VCH&utm_content=wesele&utm_campaign=weselevch).

Further, it is the author's responsibility to request permission to reproduce or adapt previously published figures. The copyright of figures generally belongs to the publisher of the journal where they first appeared. This also applies to your own previous publications. If you have not obtained permission to reproduce the figures yet, please do so before you submit your revised manuscript. We can only proceed with the publication once you have received all necessary permissions.

To submit your revision, go to <https://www.editorialmanager.com/advgenet/> and log in as an Author using your username (*****) and password. Your submission can be found under the menu item "Submissions Needing Revision". The changes to your manuscript should be highlighted in a different color in the primary "Revised Manuscript" file.

Please provide a point-by-point response letter addressed to the reviewers, including a list of changes made and a rebuttal to any comments with which you disagree. You may copy the letter into the "Respond to Reviewers" box (if it is plain text only) or upload it as a "Response Letter to Reviewers" item (if it contains figures, tables, or special formatting such as formulas or equations). If necessary, you may also upload a separate revision cover letter addressed to the editor with any other information not intended for reviewers as a "Cover Letter to Editor" item. You will also be asked to upload a .zip archive containing the production data that will be used if your manuscript is accepted. See below for more details.

We should receive your revised manuscript by 29 Jul 2022. When we receive your revised manuscript, its suitability for publication in *Advanced Genetics* will be reassessed.

We recognize that authors are doing their best to revise manuscripts under challenging circumstances due to the COVID-19 pandemic. Should you need extra time, do not hesitate to contact the editorial office.

Yours sincerely,

Kerstin Brachhold

P.S. Please help avoid delays by referring to the Manuscript Preparation Checklist (<http://www.advgenet.com/authorguidelines>) and use the appropriate article template when preparing your revised manuscript. Please also follow the instructions to prepare and upload your Production Data materials. These include: the full, non-highlighted text of your manuscript (with all figures with a resolution of at least 300 dpi, schemes, and tables) in editable format - Word DOC/X or LaTeX; a short summary (50-60 words) and an eye-catching color image for the Journal's Table of Contents; and, if applicable, Supporting Information*.

*The Supporting Information document(s) will be published alongside your article and should be non-highlighted and ready for publication. Video formats may also be included.

Copyright information should be included in each figure caption as follows (the reference number [REF] should be superscripted):

Reproduced (Adapted) with permission.[REF] Copyright YEAR, Publisher Name.

Or, for figures reproduced or adapted from CC-BY open access publications (insert the name of the specific CC-BY license for XXX; permission should be obtained from the Copyright Holder for CC-BY-NC licenses):

Reproduced (Adapted) under the terms of the XXX license.[REF] Copyright YEAR, Copyright Holder

Name(s).

REVIEWER REPORT:

Please note that reviewers may not be numbered consecutively. Where reviewers have provided additional files, these are available here: *****

EVALUATION:

Reviewer's Responses to Questions

Is the topic timely and appropriate for the research community?

Reviewer #1: Yes

Reviewer #2: Yes

Does the manuscript provide a balanced insightful view?

Reviewer #1: (No Response)

Reviewer #2: Mostly

Are the ideas presented reasonable with respect to the supporting literature?

Reviewer #1: Yes

Reviewer #2: Mostly

Which aspects of scholarly presentation require improvement (if any)?

Reviewer #1:

*Writing style

*Manuscript structure

Reviewer #2:

*Language

COMMENTS TO AUTHOR:

Reviewer #1: I read with interest the paper of Wei-Sheng Lin. I found it clear in the explanations of putative mechanisms of MD in children.

However, in my opinion:

If the paper would be an update in the movement disorders in children, it should start from a simple classification (that reflects the epidemiology of MD in children): isolated dystonia (DYT1, and emerging KMT2b), combined dystonia (ok myoclonus dystonia) and paroxysmal dystonia. Starting from phenomenology and epidemiology in terms of incidence in the paediatric population could be a great starting point.

Explanation of putative mechanisms involved are clearly reviewed, just add a paragraph on isolated dystonia.

Minor:

english and typo errors.

Reviewer #2: The article is well-written, but there are some improvements that could be made to enhance its quality.

1. The review's title is not reflective of its content. The author discussed paroxysmal dyskinesia and myoclonus-dystonia syndrome, but not all pediatric movement disorders.
2. The article should be reviewed by a native English speaker because it is full of typos.
3. There is a missing symbol in all OMIM entries: * or #
4. The phenotype of ATP1A3-related disorders is significantly more extensive than what the author described on page 8.

--

Dr. Kerstin Brachhold, Editor-in-Chief

Advanced Genetics
E-mail: AdvGenet@wiley.com
Tel: +49(0)6201-606-531

<http://www.advgenet.com>

Authors' Response to 1st Review

20 August 2022

I thank reviewers for the detailed review and comments, which is responded point by point below.

Reviewer #1: I read with interest the paper of Wei-Sheng Lin. I found it clear in the explanations of putative mechanisms of MD in children.

However, in my opinion:

If the paper would be an update in the movement disorders in children, it should start from a simple classification (that reflects the epidemiology of MD in children): isolated dystonia (DYT1, and emerging KMT2b), combined dystonia (ok myoclonus dystonia) and paroxysmal dystonia. Starting from phenomenology and epidemiology in terms of incidence in the paediatric population could be a great starting point.

Explanation of putative mechanisms involved are clearly reviewed, just add a paragraph on isolated dystonia.

Response: A brief introduction to the phenomenological classification of dystonia is added in the revised manuscript (in section 1. Introduction) as suggested by the reviewer. A brief discussion of isolated dystonia is also added in the revised manuscript (in section 4.1.), with emphasis on its relevance to the increasingly recognized role of the cerebellar circuits in pediatric movement disorders. I did not address the epidemiology because this is less relevant to the main theme of this Perspective article. Indeed, the epidemiology of pediatric movement disorders is less well characterized in the literature, and it likely varies across different geographic regions and ethnic groups.

Minor:

english and typo errors.

Response: The manuscript has been revised with the help of a colleague with extensive experience of medical English writing.

Reviewer #2: The article is well-written, but there are some improvements that could be made to enhance its quality.

1. The review's title is not reflective of its content. The author discussed paroxysmal dyskinesia and myoclonus-dystonia syndrome, but not all pediatric movement disorders.

Response: This Perspective article focuses on several movement disorders primarily featuring dystonia, but it also touches upon other forms of involuntary movements (e.g., choreoathetosis in paroxysmal dyskinesia, and myoclonus in myoclonus-dystonia syndrome). Moreover, the discussions in section 4 are broadly applicable to other pediatric movement disorders. A subtitle (“Lessons learned from genetic dystonias and related disorders”) is added to better reflect the content of this Perspective article.

2. The article should be reviewed by a native English speaker because it is full of typos.

Response: The revised paper has been reviewed by an anonymous colleague with extensive experience in medical English writing.

3. There is a missing symbol in all OMIM entries: * or #

Response: The missing symbol in OMIM entries was added in the revised manuscript.

4. The phenotype of ATP1A3-related disorders is significantly more extensive than what the author described on page 8.

Response: I concur with the reviewer’s opinion that ATP1A3-related disorders have a broad phenotypic spectrum, including both paroxysmal and progressive forms and both epileptic and nonepileptic events, as tabulated in an editorial (<https://doi.org/10.1016/j.ejpn.2019.05.007>) and reviewed elsewhere. I did not mention the details just because my point here is the phenotypic similarities between some ATP1A3- and PRRT2-related disorders, presumably reflecting disturbed molecular interactions between ATP1A3 and PRRT2. The sentence has been rewritten as below to avoid confusion:

ATP1A3 is associated with a wide spectrum of neurological phenotypes which are sometimes complicated, and both *PRRT2* and *ATP1A3* mutations are implicated in distinct forms of paroxysmal dyskinesia and epilepsy. The phenotypic similarities suggest that interference of their interactions may be related to the pathogenesis of these disorders.

Dear Professor Lin,

Thank you for submitting your revised manuscript entitled "Translating genetic discovery into mechanistic understanding of pediatric movement disorders: Lessons from genetic dystonias and related disorders" (Perspective, No. ggn2.202200018R1) to Advanced Genetics.

We are now satisfied with the scientific content of this Review. Your manuscript, however, still requires substantial revision to improve the overall quality of the writing. It is important that Reviews published at Advanced Genetics are engagingly written and accessible to a broad audience of readers.

We therefore suggest that you ask a native English speaker or a professional editing service to assist you with improving the text throughout your manuscript. The manuscript will need editing for clarity and logical flow, not just basic grammar. To help with this process you may wish to contact a professional language editing service, such as Wiley Editing Services (https://wileyeditingservices.com/en/article-preparation/english-language-editing?utm_source=email&utm_medium=referral&utm_term=VCH&utm_content=wesele&utm_campaign=weselevch). Please note that without significant language improvement your manuscript unfortunately will not be acceptable for publication.

To submit your revision, go to <https://www.editorialmanager.com/advgenet/> and log in as an Author using your username (*****) and password. Your submission can be found under the menu item "Submissions Needing Revision". The changes to your manuscript should be highlighted in a different color in the primary "Revised Manuscript" file. You may upload your revision cover letter addressed to the editor as a "Cover Letter to Editor" item, including a list of changes made. You will also be asked to upload a .zip archive containing the production data that will be used if your manuscript is accepted. See below for more details.

We should receive your revised manuscript by 25 Sep 2022. Once we receive your revised manuscript, we will provide a final decision as soon as possible. Should you need extra time, do not hesitate to contact the editorial office.

Yours sincerely,

Andrew Hufton

--

Dr Andrew Hufton, Editor
Advanced Genetics
E-mail: AdvGenet@wiley.com
Tel: +49(0)6201-606-362

<http://www.advgenet.com>

Authors' Response to 2nd Decision

27 September 2022

Comments:

We are now satisfied with the scientific content of this Review. Your manuscript, however, still requires substantial revision to improve the overall quality of the writing. It is important that Reviews published at Advanced Genetics are engagingly written and accessible to a broad audience of readers.

Response:

With the help of Wiley Editing Services, we have thoroughly revised the manuscript to improve the clarity and logical flow.

Final Decision

27 September 2022

Dear Professor Lin,

Thank you for submitting your manuscript entitled "Translating genetic discovery into a mechanistic understanding of pediatric movement disorders: Lessons from genetic dystonias and related disorders" (Perspective, No. ggn2.202200018R2) to Advanced Genetics.

I'm pleased to inform you that your manuscript has been accepted for publication without change.

We will copyedit the accepted version of your manuscript and if we require any further information at this stage we will contact you. After copyediting we will let you know when you can expect to receive the proofs. Instructions for returning your proof corrections will be provided when the proofs are sent to you.

Further, it is the author's responsibility to request permission to reproduce or adapt previously published figures. The copyright of figures generally belongs to the publisher of the journal where they first appeared. This also applies to your own previous publications. If you have not obtained permission to reproduce the figures yet, please do so immediately. We can only proceed with the publication once you have received all necessary permissions.

All articles published in Advanced Genetics are fully open access: immediately and freely available to read, download and share. Advanced Genetics charges a publication fee to cover publication costs. The corresponding author for this manuscript should have already received a quote with the article publication fee, and will soon receive an e-mail invitation to register with or log in to Wiley Author Services (<https://authorservices.wiley.com>). After logging into Wiley Author Services, the publication fee can be paid by credit card, or an invoice or pro forma can be requested. Payment of the publication charge must be received before the article will be published online.

Thank you for choosing Advanced Genetics for publishing your work. I hope you will consider us for the publication of your future manuscripts.

Yours sincerely,

Andrew Hufton

P.S.: If you believe your images might be appropriate for use on the cover of Advanced Genetics, and you would like your paper to be considered for the cover, please e-mail us your layout suggestions with a short description. For details on cover image preparation, please see the cover gallery on <http://www.advgenet.com>.

--

Dr Andrew Hufton, Editor
Advanced Genetics
E-mail: AdvGenet@wiley.com
Tel: +49(0)6201-606-362

<http://www.advgenet.com>